# Fasting Glucose Level on the Oral Glucose Tolerance Test Is Associated with the Need for Pharmacotherapy in Gestational Diabetes Mellitus

**DOI:** 10.3390/nu15051226

**Published:** 2023-02-28

**Authors:** Natassia Rodrigo, Deborah Randall, Farah Abu Al-Hial, Kathleen L. M. Pak, Alexander Junmo Kim, Sarah J. Glastras

**Affiliations:** 1Department of Diabetes, Endocrinology and Metabolism, Royal North Shore Hospital, St Leonard’s, NSW 2065, Australia; 2North Precinct, Faculty of Medicine and Health, The University of Sydney, Camperdown, NSW 2006, Australia; 3Kolling Institute of Medical Research, Northern Sydney Local Health District, St Leonard’s, NSW 2065, Australia; 4Women’s and Babies Research, Northern Clinical School, The University of Sydney, St Leonard’s, NSW 2065, Australia

**Keywords:** gestational diabetes mellitus, fasting hyperglycaemia, pharmacotherapy

## Abstract

Gestational diabetes mellitus (GDM) has a rapidly increasing prevalence, which poses challenges to obstetric care and service provision, with known serious long-term impacts on the metabolic health of the mother and the affected offspring. The aim of this study was to evaluate the association between glucose levels on the 75 g oral glucose tolerance test and GDM treatment and outcomes. We performed a retrospective cohort study of women with GDM attending a tertiary Australian hospital obstetric clinic between 2013 and 2017, investigating the relationship between the 75 g oral glucose tolerance test (OGTT) glucose values, and obstetric (timing of delivery, caesarean section, preterm birth, preeclampsia), and neonatal (hypoglycaemia, jaundice, respiratory distress and NICU admission) outcomes. This time frame encompassed a change in diagnostic criteria for gestational diabetes, due to changes in international consensus guidelines. Our results showed that, based on the diagnostic 75 g OGTT, fasting hyperglycaemia, either alone or in combination with elevated 1 or 2 h glucose levels, was associated with the need for pharmacotherapy with either metformin and/or insulin (*p* < 0.0001; HR 4.02, 95% CI 2.88–5.61), as compared to women with isolated hyperglycaemia at the 1 or 2 h post-glucose load timepoints. Fasting hyperglycaemia on the OGTT was more likely in women with higher BMI (*p* < 0.0001). There was an increased risk of early term birth in women with mixed fasting and post-glucose hyperglycaemia (adjusted HR 1.72, 95% CI 1.09–2.71). There were no significant differences in rates of neonatal complications such as macrosomia or NICU admission. Fasting hyperglycaemia, either alone or in combination with post-glucose elevations on the OGTT, is a strong indicator of the need for pharmacotherapy in pregnant women with GDM, with significant ramifications for obstetric interventions and their timing.

## 1. Introduction

Gestational diabetes mellitus (GDM), defined as any degree of glucose intolerance with onset or first recognition during pregnancy [1], has become the fastest growing subtype of diabetes in many countries including Australia, with a doubling of incidence over the past decade, to now encompass at least 14% of all Australian pregnancies [2,3]. The rise in incidence is multi-factorial, with increasing maternal age and a higher prevalence of obesity central to these trends [4]. The changes to the diagnostic criteria are also implicated in the increased incidence of GDM. Since the gradual uptake of the GDM diagnostic criteria recommended by the International Association of the Diabetes and Pregnancy Study Groups (IADPSG), the incidence of GDM in Australia has more than doubled. There are some benefits of diagnosing and treating GDM such as a reduction in rates of macrosomia, however, there is also a trend towards earlier timing of birth in mothers with GDM [2,5].

Though the treatment of hyperglycaemia in pregnancy improves pregnancy-related outcomes, there is limited evidence to guide decisions about how best to adapt to the demands of the growing patient population [6]. The increasing number of women with GDM necessitates the optimisation of treatment pathways, to provide greater medical and obstetric surveillance and intervention to the women most in need, whilst reducing unnecessary intervention in women with mild GDM. Women with GDM who require pharmacotherapy, such as insulin or metformin, have higher rates of adverse pregnancy outcomes [7]. Therefore, earlier triage of patients with GDM into treatment pathways based on the likelihood of requiring pharmacotherapy may optimise patient models of care. 

Currently, the results of the universal 75 g oral glucose tolerance test (OGTT) performed at 24–28 weeks’ gestation provide important information that could be used to stratify women into low- and high-risk models of care. The aim of this present study was to assess the maternal characteristics and perinatal outcomes of women attending a tertiary multidisciplinary antenatal clinic, according to the OGTT result diagnostic of GDM. We hypothesised that women with higher glucose values on OGTT, or multiple glucose levels above the diagnostic target, would be more likely to need pharmacotherapy, and have adverse maternal and neonatal outcomes. 

## 2. Materials and Methods

### 2.1. Study Design

A retrospective, cohort study was performed by reviewing the electronic medical record of women with GDM, who attended the multi-disciplinary Specialist Obstetric Clinic at Royal North Shore Hospital, Sydney, Australia from 2013 to 2017. Approval for this study was obtained from the Northern Sydney Local Health District Research Ethics Committee (Study Reference Number RESP/15/107). This study followed the precepts delineated in the “Strengthening the Reporting of Observational Studies in the Epidemiology (STROBE)” statement for cohort studies [8].

Maternal demographics were collected including age, ethnicity, body mass index (BMI) recorded at the first visit to the antenatal clinic, antenatal history, past medical history of polycystic ovarian syndrome (PCOS) and family history of diabetes or hypertension.

Gestational age at GDM diagnosis and diagnostic values of the 75 g oral glucose tolerance test (OGTT) results were recorded. GDM was diagnosed in accordance with the Australian Diabetes in Pregnancy Society (ADIPS) guidelines [9], which were based on the 75 g OGTT. Over the course of the study, ADIPS diagnostic criteria shifted due to the IADPSG recommendations, reflected in this study period. The old criteria involved an initial glucose challenge using 50 g of glucose, with a 1 h threshold of ≥7.8 mmol/L triggering the need for a 75 g OGTT. Criteria for diagnosis were as follows: fasting plasma glucose ≥5.5 mmol/L or 2 h plasma glucose ≥7.8 mmol/L. This was recommended till the end of December 2014, with the new ADIPS diagnostic criteria introduced in January 2015; the 50 g challenge test was removed and the 75 g OGTT was recommended for all patients with the following diagnostic criteria: fasting plasma glucose ≥5.1 mmol/L, 1 h plasma glucose ≥10 mmol/L or 2 h plasma glucose ≥8.5 mmol/L. 

In our tertiary clinic, all women with GDM are managed by a multidisciplinary team, commencing with a standardised group GDM education session, led by a credentialed diabetes educator and a dietitian. Women were instructed to undertake self-monitoring of blood glucose (SMBG), perform fasting and 2 h postprandial glucose (PPG) levels and apply a low fat, carbohydrate portioned diet (total 175 g daily). In women not meeting treatment targets (until December 2014: fasting <5.5 mmol/L, PPG < 6.7 mmol/L, and after January 2015: fasting <5.0 mmol/L, PPG < 6.7 mmol/L according to ADIPS guideline recommendations), treatment was intensified to include pharmacotherapy. Therefore, in this study, the diet group comprised of women not requiring pharmacotherapy, the metformin group comprised women treated with metformin, to a maximum dose of 2 g daily, and the insulin group was treated with insulin monotherapy (short-acting: aspart, long-acting: isophane or detemir), and the metformin/insulin group was treated with both metformin and insulin. 

Foetal growth ultrasound data (estimated foetal weight, abdominal circumference), perinatal outcomes, namely gestational age at delivery, mode of delivery (normal vaginal delivery (NVB), instrumental, including forceps and vacuum delivery, planned or emergency lower segment caesarean section (LSCS)), gender, birth weight, large for gestational age (LGA: defined as >90th centile for gender and gestational age) [10], small for gestational age (SGA: defined as <10th centile for gender and gestational age) [10] neonatal hypoglycaemia, jaundice, respiratory distress, neonatal intensive care unit (NICU) admission and the incidence of hypertensive conditions in pregnancy were recorded from the medical record.

### 2.2. Statistical Analysis

Differences between groups were compared using one-way analysis of variance (ANOVA) for continuous data. Categorical data were analysed using chi-squared test and logistic regression. The association between the 75 g OGTT glucose measures (fasting, 1 and 2 h) and the time to medication (either insulin, metformin or both) was modelled using Cox regression with gestational age as the time scale, with entry on the gestational day of GDM diagnosis. The outcome was gestational day at first treatment, and women with diet-controlled GDM were censored at the day of birth. The model was adjusted for ethnicity (Caucasian, South Asian, South-East Asian, Other), birth year (in year categories), parity (1, 2, 3+), history of PCOS, maternal age (entered as a continuous variable centred at 34), BMI (entered as a continuous variable centred at 24 Kg/m^2^). The interaction between the glucose measures and BMI (grouped as <25 and 25+) was also investigated, adjusting for the same covariates as above, without continuous BMI. Additionally, a composite glucose variable was created, segmenting the cohort into 6 groups based on their results on all 3 glucose measures (fasting and either 1 h or 2 h criteria, fasting criteria only, 1 h and 2 h criteria but not fasting, 1 h criteria only, 2 h criteria only, other), with criteria based on the IADPSG diagnostic levels.

The association between the 75 g OGTT glucose measures and perinatal outcomes were investigated using a Fine–Gray competing risk model [11]. A separate model was run for each perinatal outcome (preterm birth <37 weeks, early term birth <39 weeks, macrosomia >4 kg, large for gestational age, small for gestational age, birth by caesarean section (CS), birth by CS or instrumental, gestational hypertension at birth, transfer to NICU, neonatal hypoglycaemia, newborn respiratory distress). Birth, without the specified outcome, was considered a competing risk event. Time at risk of the outcomes started at 24 weeks’ gestation as the cohort was not at risk of the outcomes (measured at birth) until they reached at least 24 weeks’ gestation. The models are presented unadjusted, adjusted for ethnicity, birth year, parity, maternal age and BMI (adjusted model 1), and additionally adjusted for treatment mode (diet, insulin, metformin and both insulin and metformin; adjusted model 2).

The regression models produced hazard ratios (HRs) and 95% confidence intervals, with the hazard ratio a measure of the relative risk of an outcome for the next time period for those yet to experience the outcome.

Statistical analyses were carried out using SAS Version 9.4 (SAS Institute Inc., Cary, NC, USA). *p*-value < 0.05 was considered statistically significant.

## 3. Results

### 3.1. Study Population

Between 2013 and 2017, 654 singleton pregnancies were recorded, with mean maternal age of 34.1 years, parity of 1.6 and early pregnancy BMI of 25.4 kg/m^2^. The mean gestational age of GDM diagnosis was 25.6 weeks. There was no difference in maternal age or parity between the group diagnosed with GDM by the old criteria, compared to those diagnosed by the new criteria. However, women diagnosed with the new criteria had a higher early pregnancy BMI (25.9 vs. 24.6 kg/m^2^, Table 1). Women diagnosed with GDM by the new versus old criteria had higher fasting and 1 h glucose values on OGTT, however lower 2 h glucose value, compared to women diagnosed with GDM by the old criteria (4.7 vs. 4.6, 9.6 vs. 9.3 and 8.3 vs. 8.6 mmol/L at 0, 1, 2 h timepoint on 75 g OGTT).

Women were categorised into diagnostic groups based on the OGTT results (Table 2). Most women were diagnosed with GDM on one diagnostic timepoint alone (13.9%, 11.9% and 45.1% on a single elevated result at 0, 1 or 2 h timepoint, respectively). There was a change in the proportion of women in each diagnostic group over time (2013–2017); from 2015, there was a significant increase in women diagnosed on the 1 h glucose value, corresponding with the changed diagnostic criteria to include a 1 h glucose value. The mean maternal age and parity were similar across all GDM diagnostic groups (Table 2). A greater proportion of women with low–normal BMI (<25 kg/m^2^) met the GDM diagnostic criteria based on the 1 h and/or 2 h glucose tests, whereas women with high BMI > 30 kg/m^2^ were more likely to have the GDM diagnosis based on fasting glucose with or without elevated 1 h and 2 h glucose values (*p* < 0.0001). With regards to ethnicity, there was a high proportion of South-East Asian women within groups diagnosed by the 1 h and/or 2 h glucose result. There was a higher representation of Caucasian ethnicity in the group of women diagnosed with GDM by elevated fasting glucose. There was no difference in the proportion of women with a history of GDM, pre-eclampsia, PCOS or a family history of diabetes between combined diagnostic groups.

### 3.2. GDM Diagnostic Group and Treatment Differences

Within this cohort of women with GDM, one in two women managed throughout pregnancy with lifestyle intervention alone (Table 3). In women who required pharmacotherapy, the mean age at the time of intervention was 27.6 weeks. Insulin was the most commonly prescribed medication used to manage GDM, 46.9% of women were commenced on therapy within the 28–32-week gestational period, with 14.5% commencing treatment at 33 weeks’ gestation or later.

Women managed by lifestyle intervention (diet) alone had lower fasting glucose levels on the 75 g OGTT compared to women who required treatment (*p* < 0.001, Table 3). In women with fasting glucose levels <4.3 mmol/L on the OGTT, 75.4% were managed with diet intervention alone. In contrast, 73.6% of women with a fasting glucose above 5.1 mmol/L required pharmacotherapy.

The 1 h glucose level on the diagnostic 75 g OGTT was predictive of the need for treatment (*p* = 0.016, Table 3). Two-thirds of women with 1 h glucose levels greater than 10.8 mmol/L were managed with pharmacotherapy, and specifically, 75.1% of women were treated with insulin, mostly without metformin. Similarly, there were significant differences between the 2 h glucose level on the 75 g OGTT and the need for pharmacotherapy. A total of 36.7% of women with a 2 h glucose level of less than 8.0 mmol/L were managed on diet alone, whereas 8.9% required both metformin and insulin therapy. A total of 48.5% of women with a glucose level of 9.4 mmol/L or more were managed with diet alone, and 8.3% required both metformin and insulin therapy.

Almost 50% of the cohort were diagnosed with GDM based on an elevated fasting glucose level with or without an elevated 1 h or 2 h glucose level. Interestingly, of the women with a diagnostic 2 h glucose level alone, 63.7% were able to manage with diet intervention alone, whereas 27.3% of women with a diagnostic fasting glucose level alone managed with diet alone. Women with ≥ 2 glucose values exceeding the diagnostic targets were the most likely to require insulin therapy, with 63.8% of women with fasting and 1 h or 2 h readings meeting the diagnostic threshold requiring insulin therapy. A total of 34.5% of women with diagnostic 1 and 2 h readings required insulin therapy (*p* < 0.0001).

On the diagnostic 75 g OGTT, women with a higher fasting glucose level were 1.77 times more likely to need medication than women with a fasting glucose level was less than 4.3 mmol/L (Figure 1, Appendix A). Women with a fasting level of 5.4 mmol/L or more were six times more likely to need medication than those with fasting levels less than 4.3 mmol/L (CI 3.76–9.18). After adjusting for ethnicity, birth year, parity, maternal age and BMI, women with a fasting glucose value ≥ 5.4 mmol/L were still 3.9 times more likely to need pharmacotherapy than women with a fasting glucose < 4.3 mmol/L (CI 2.41–6.31). Women with fasting levels between 4.6–5.1 mmol/L had almost three times the risk of needing pharmacotherapy (95% CI of 1.93–4.33 unadjusted). Following adjustment for covariates this was still significant, with a hazard ratio of 2.62 (CI 1.74–3.95).

The risk of needing pharmacotherapy was analysed by combined groups (Figure 1, Appendix A). Overall, the group with fasting hyperglycaemia alone (≥5.1 mmol/L) had twice the likelihood of needing treatment (CI 1.37–3.14). Similarly, women with a 1 h OGTT reading of 10.8 mmol/L or more were 2.15 more likely to require treatment after adjustment. An elevated 2 h glucose reading alone did not increase the risk of requiring treatment, however, a combination of 1 and 2 h readings above the diagnostic cut-offs (and no fasting hyperglycaemia) had 1.82 times the hazard of needing pharmacotherapy (CI 1.30–2.56), and combined fasting and post glucose hyperglycaemia had an adjusted hazard ratio of 2.96 (CI 2.02–4.34) (Appendix A).

Women with lower BMI had a lower risk of pharmacotherapy than women with higher BMI across all glucose categories and groups (Figure 2a–d). There was no significant interaction between BMI groups (≥25 kg/m^2^ or <25 kg/m^2^) and the OGTT glucose values.

### 3.3. GDM Diagnostic Group and Third Trimester Foetal Ultrasound

Within the total cohort of women, 58% had a foetal ultrasound performed in the third trimester. There were no significant differences in foetal weight (EFW) or abdominal circumference (AC) by the third trimester scan results, between the diagnostic groups based on the OGTT results (Appendix A). This may be due to the differences in gestational age at which the scans were carried out on average, which was significantly different between groups (*p* < 0.05). Further, women with diet-controlled GDM may not routinely be offered additional growth scans.

### 3.4. GDM Diagnostic Group and Perinatal Outcomes

The gestational age at delivery varied according to GDM diagnostic group, though within all diagnostic groups, the highest proportion of women delivered within the 39th week. (*p* = 0.031, Table 4). There was a trend towards earlier delivery in women with ≥2 glucose values diagnostic of GDM (either fasting and 1 h and 2 h, or 1 h and 2 h only groups), with 57.2% of women diagnosed with GDM by combined fasting and post glucose levels giving birth before 39 weeks.

The majority of women (64.2%) had a vaginal birth (49.2% normal vaginal, 15% instrumental), whilst 30.4% had a planned caesarean section and 5.2% had an emergency caesarean section.

There was no significant difference in the mode of delivery between groups (normal vaginal delivery rate was 49.2% overall), LGA (9.6%), SGA (11%), macrosomia (6%), pre-eclampsia occurred (1.7%) and gestational hypertension (5%). Further, there was no significant difference in neonatal complications, including respiratory distress (7.6%), jaundice (7.3%) or neonatal hypoglycaemia (8.4%) between the groups (Table 4).

As detailed in the statistical methods, three statistical models were utilised to determine the impact of the diagnostic group on perinatal outcomes, with each model using the fasting alone group as the reference category: (1) unadjusted model, (2) model 1 adjusted for ethnicity, birth year, parity, maternal age and BMI and (3) model 2 adjusted for ethnicity, birth year, parity, maternal age, BMI and treatment type. Both unadjusted and adjusted models demonstrated no significant association between the diagnostic groups on OGTT and preterm birth less than 37 weeks (Figure 3, Appendix A). There was an increased risk of early term birth (<39 weeks) in women with mixed fasting and post-glucose hyperglycaemia (unadjusted HR 1.74 (CI 1.13–2.67)), and it remained significant after adjustment in model 1 (HR1.72 (CI 1.09–2.71)) and model 2 (1.65 (CI 1.06–2.56)). Neither the unadjusted model, model 1 nor model 2 found a relationship between the diagnostic group and macrosomia, large for gestational age, or small for gestational age. Caesarean section was more likely in the maternal group when the diagnostic group included both 1 and 2 h, but not fasting hyperglycaemia, in model 1 (HR 1.85 (CI 1.07–3.19)) and model 2 (HR 1.86 (CI 1.08–3.23)), as well as for 1 h only in model 1 (HR 1.85 (CI 1.05–3.27)) and model 2 (HR 1.91 (CI 1.08–3.40)).

There were no significant differences between the GDM diagnostic category and other neonatal outcomes such as neonatal ICU, hypoglycaemia or respiratory distress.

## 4. Discussion

In this large, single-centre retrospective study, we determined that women with fasting hyperglycaemia ≥ 5.1 mmol/L, at the time of the diagnostic OGTT were more likely to need pharmacotherapy for the treatment of GDM than those with lower fasting levels. Women with fasting levels of OGTT between 4.6–5.1 mmol/L (lower than the current GDM diagnostic thresholds) were still at risk of needing medication. Women with the GDM diagnosis based on elevated 1 h or 2 h glucose levels were more likely to maintain diet intervention alone throughout the pregnancy than those with fasting hyperglycaemia. The 1 h glucose level was associated with a higher risk of treatment, but only at the very highest level of 10.8 mmol/L or more. Almost 50% of women were diagnosed with GDM by the 2 h glucose value on OGTT alone, yet the 2 h value had the least association with the need for pharmacotherapy or maternal and foetal outcomes.

GDM arises from a multifaceted process of β-cell dysfunction, oxidative stress and chronic insulin resistance, which is compounded by placental metabolic hormones and cytokines, thereby transforming the physiological insulin resistance of pregnancy into the pathophysiological condition of GDM [12]. Post-prandial glucose levels can be manipulated by lifestyle modification [13,14], including dietary adjustment and exercise, strongly advocated by diabetes education teams. Fasting hyperglycaemia occurs as a result of diurnal changes in cortisol and growth hormone secretion and increased hepatic glucose production and reduced hepatic insulin sensitivity amplified in pregnancy [15,16,17]. Lifestyle manipulation is unlikely to alter these physiological drivers and is rarely sufficient to overcome persistent fasting hyperglycaemia resulting in the need for pharmacotherapy intervention, as elucidated in this study. Furthermore, women with both fasting and post-prandial hyperglycaemia are likely to have multiple pathways of metabolic dysregulation at play, thereby making pharmacotherapy highly likely to aid the control of hyperglycaemia in this group of women with GDM [17,18].

In this study, it was reassuring that the glucose levels on OGTT were not associated with the majority of maternal and neonatal outcomes studied, suggesting successful treatment to target glucose levels. Women with GDM have demonstrated higher risks of pre-eclampsia, shoulder dystocia, caesarean section, LGA and malformations [19,20]. As demonstrated by the Australian Carbohydrate Intolerance Study in Pregnancy Women (ACHOIS) Trial Group, women treated with dietary advice, blood glucose monitoring and pharmacotherapy had significantly lower rates of serious perinatal complications [6]. Treatment benefit has been further supported by many other studies [21,22]. The diagnosis of GDM has been found to be associated with increased intervention, such as caesarean section [23,24]. Compared to the national average, in which 64% of women had a vaginal birth in 2019 [25], reassuringly in our study, 64.2% of women with GDM had vaginal births, either normal or instrumental.

The results of this study provide important information to guide clinicians and policymakers on ways to triage women to appropriate GDM models of care. Because women with ≥2 glucose levels diagnostic of GDM and/or fasting glucose value ≥4.6 mmol/L were more likely to need medication, these women could be triaged into high-risk antenatal clinics, during which prescribers are present to initiate and titrate pharmacotherapy, and closely monitor the pregnancy. In contrast, women with GDM diagnosed by either 1 h or 2 h glucose level alone, and glucose level <4.6 mmol/L could be allocated to lower risk models of care, given that these women are less likely to require pharmacotherapy intervention, with good maternal and perinatal outcomes. Women with high BMI are more likely to need medication, and hence these women may require closer monitoring. Together, these factors may have the potential to contribute to more personalised risk calculations to inform the models of care most relevant to an individual woman.

A limitation of this study is that it was a single-centre study. However, the centre is a tertiary referral centre, with a diverse and populous catchment. Additionally, the retrospective nature of this study meant that some data, such as ultrasound findings, were not available for the entire group of subjects. Further, this was not a randomised study, and therefore therapy decisions were subject to patient and/or physician preference. The changing diagnostic criteria, over the course of the study period also introduced heterogeneity to the data, although we did base our study groups on the current OGTT diagnostic criteria.

## 5. Conclusions

In summary, this study demonstrates that fasting hyperglycaemia on the OGTT at the time of GDM diagnosis is strongly associated with the need for pharmacotherapy. GDM is a condition where timely treatment is imperative, and attempting diet and lifestyle modification is costly, both in time and resources. Our study provides data to inform clinical triage of patients most likely to require pharmacotherapy from the time of GDM diagnosis, and it informs models of care and treatment pathways that can streamline finite resources to the women most likely to benefit from pharmacotherapy and closer obstetric monitoring.

## Figures and Tables

**Figure 1 nutrients-15-01226-f001:**
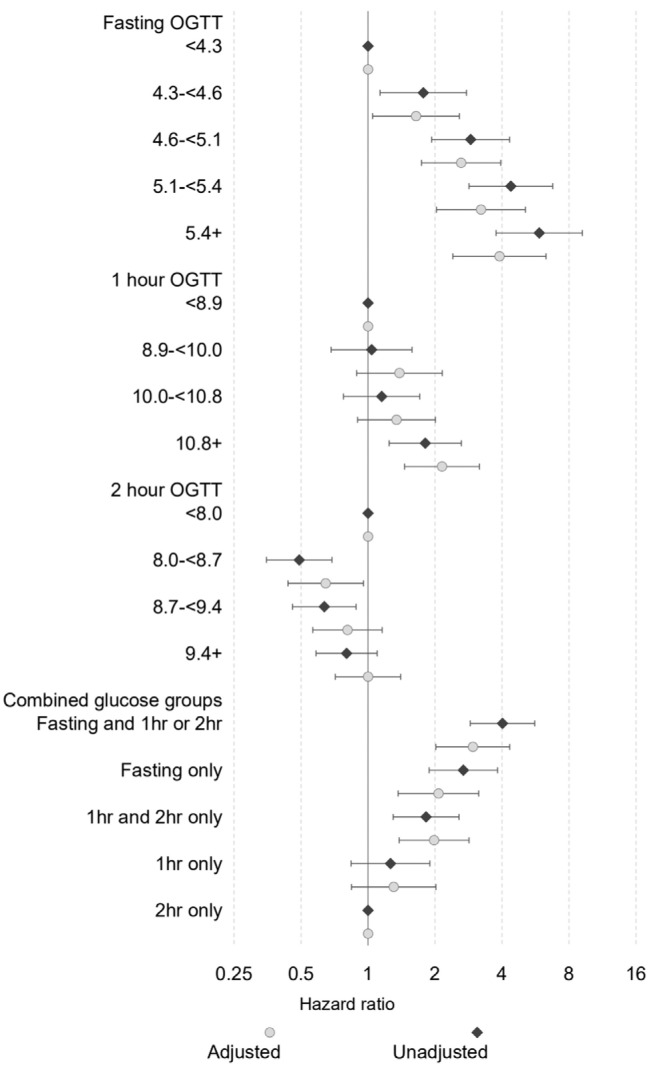
Hazard ratio for medication by glucose tolerance test result categories.

**Figure 2 nutrients-15-01226-f002:**
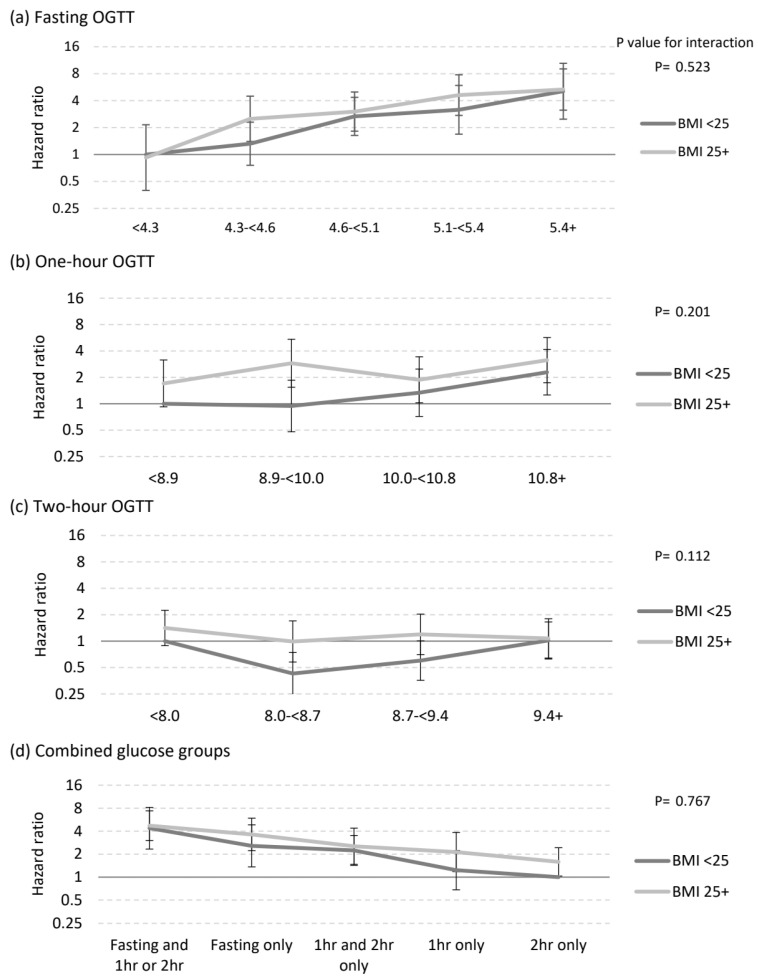
Adjusted hazard ratio for needing medication (metformin and/or insulin) by glucose results and BMI category for (**a**) fasting glucose reading on OGTT, (**b**) one hour glucose reading on OGTT, (**c**) 2 h glucose reading on OGTT, (**d**) combined glucose groups. Adjusted for ethnicity (Caucasian, South Asian, South-East Asian, Other), birth year (in categories), parity (1, 2, 3+), maternal age (centred continuous). OGTT, oral glucose tolerance test.

**Figure 3 nutrients-15-01226-f003:**
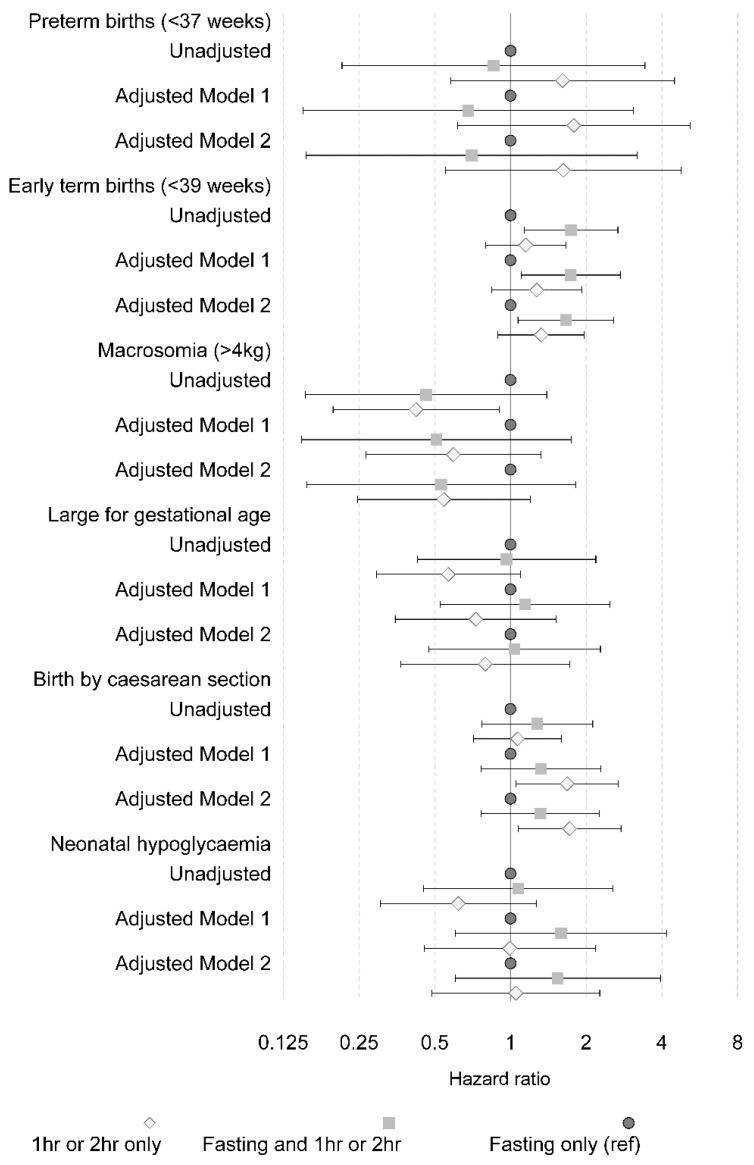
Representation of outcomes against glucose levels as a forest plot. Adjusted for ethnicity (Caucasian, South Asian, South-East Asian, Other), birth year (in categories), parity (1, 2, 3+), maternal age (centred continuous), BMI (centred continuous). Adjusted as per model 1 and additionally including treatment type (diet, insulin, metformin, insulin and metformin).

**Table 1 nutrients-15-01226-t001:** Baseline demographics.

	Old Criteria		New Criteria		*p* Value
	Mean ± SD	Missing *n* (%)	Mean ± SD	Missing *n* (%)	
Maternal age (weeks)	33.9± 4.3	0	34.1 ± 4.5	0	0.6
Parity	1.6 ± 0.8	0	1.7 ± 0.8	0	0.3
Early pregnancy BMI (kg/m^2^)	24.6 ± 6.0	22	25.9 ± 6.1	10	0.004
Gestational age at GDM diagnosis	26.3 ±5.8	2	25.6 ± 5.9	3	0.09
OGTT fasting (mmol/L)	4.6 ± 0.6	3	4.7 ± 0.7	3	0.01
OGTT 60 min (mmol/L)	9.3 ± 1.7	163	9.6 ± 1.7	12	0.05
OGTT 120 min (mmol/L)	8.6 ± 1.3	3	8.3 ± 1.7	4	0.004
Gestational age at GDM treatment (weeks)	28.4 ± 5.1	4	28.1 ± 5.9	1	0.6

**Table 2 nutrients-15-01226-t002:** Maternal characteristics by combined diagnostic group.

Maternal Characteristics	Total *n* (%)	Fasting and 1 h or 2 h *n* (%)	Fasting Only *n* (%)	1 h and 2 h Only *n* (%)	1 h Only *n* (%)	2 h Only *n* (%)	Chi-Square *p*-Value
Total	654 (100)	77 (11.8%)	91 (13.9%)	113 (17.3%)	78 (11.9%)	295 (45.1%)	
Year of delivery							
2013	111 (17.0)	5 (6.5)	9 (9.9)	19 (16.8)	5 (6.4)	73 (24.7)	<0.0001
2014	125 (19.1)	6 (7.8)	11 (12.1)	9 (8.0)	2 (2.6)	97 (32.9)	
2015	123 (18.8)	16 (20.8)	28 (30.8)	20 (17.7)	21 (26.9)	38 (12.9)	
2016	184 (28.1)	24 (31.2)	31 (34.1)	40 (35.4)	27 (34.6)	62 (21.0)	
2017	111 (17.0)	26 (33.8)	12 (13.2)	25 (22.1)	23 (29.5)	25 (8.5)	
Maternal age							
<25	10 (1.5)	2 (2.6)	2 (2.2)	0 (0.0)	3 (3.8)	3 (1.0)	0.478
25–34	373 (57.0)	48 (62.3)	50 (54.9)	63 (55.8)	45 (57.7)	167 (56.6)	
35+	271 (41.4)	27 (35.1)	39 (42.9)	50 (44.2)	30 (38.5)	125 (42.4)	
Parity							
1	341 (52.1)	34 (44.2)	54 (59.3)	60 (53.1)	41 (52.6)	152 (51.5)	0.645
2	233 (35.6)	30 (39.0)	26 (28.6)	42 (37.2)	26 (33.3)	109 (36.9)	
3+	80 (12.2)	13 (16.9)	11 (12.1)	11 (9.7)	11 (14.1)	34 (11.5)	
Early pregnancy BMI							
<18.5	30 (4.6)	0 (0.0)	1 (1.1)	5 (4.4)	2 (2.6)	22 (7.5)	<0.0001
18.5–24.9	322 (49.2)	23 (29.9)	23 (25.3)	65 (57.5)	41 (52.6)	170 (57.6)	
25.0–29.9	157 (24.0)	21 (27.3)	26 (28.6)	31 (27.4)	23 (29.5)	56 (19.0)	
30.0–39.9	87 (13.3)	18 (23.4)	26 (28.6)	3 (2.7)	10 (12.8)	30 (10.2)	
40+	26 (4.0)	11 (14.3)	7 (7.7)	1 (0.9)	1 (1.3)	6 (2.0)	
Missing	32 (4.9)	4 (5.2)	8 (8.8)	8 (7.1)	1 (1.3)	11 (3.7)	
Ethnicity							
Caucasian	226 (34.6)	31 (40.3)	35 (38.5)	33 (29.2)	33 (42.3)	94 (31.9)	<0.0001
South Asian	121 (18.5)	15 (19.5)	26 (28.6)	20 (17.7)	12 (15.4)	48 (16.3)	
South-East Asian	256 (39.1)	18 (23.4)	23 (25.3)	54 (47.8)	30 (38.5)	131 (44.4)	
Middle Eastern	30 (4.6)	10 (13.0)	4 (4.4)	4 (3.5)	3 (3.8)	9 (3.1)	
Other	21 (3.2)	3 (3.9)	3 (3.3)	2 (1.8)	0 (0.0)	13 (4.4)	
Previous GDM	95 (14.5)	8 (10.4)	13 (14.3)	17 (15.0)	14 (17.9)	43 (14.6)	0.768
Previous PE	20 (3.1)	2 (2.6)	4 (4.4)	1 (0.9)	4 (5.1)	9 (3.1)	0.474
Previous HT	29 (4.4)	4 (5.2)	9 (9.9)	2 (1.8)	5 (6.4)	9 (3.1)	0.034
PCOS	62 (9.5)	8 (10.4)	11 (12.1)	9 (8.0)	5 (6.4)	29 (9.8)	0.735
Family history HT	245 (37.5)	20 (26.0)	30 (33.0)	36 (31.9)	38 (48.7)	121 (41.0)	0.011
Family history DM	292 (44.6)	40 (51.9)	44 (48.4)	53 (46.9)	31 (39.7)	124 (42.0)	0.414

**Table 3 nutrients-15-01226-t003:** Glucose tolerance test results by treatment modality.

	Total	Diet	Metformin	Insulin	Metformin and Insulin	Chi-Square *p*-Value
Total *n* (%)	654 (100%)	329 (50.3%)	79 (12.1%)	194 (29.7%)	50 (7.6%)	
Fasting OGTT						
<4.3	142(100.0)	107 (75.4)	9 (6.3)	22 (15.5)	3 (2.1)	<0.001
4.3-<4.6	136 (100.0)	83 (61.0)	15 (11.0)	27 (19.9)	10 (7.4)	
4.6-<5.1	192 (100.0)	88 (45.8)	24 (12.5)	71 (37.0)	9 (4.7)	
5.1-<5.4	96 (100.0)	27 (28.1)	20 (20.8)	33 (34.4)	16 (16.7)	
5.4+	82 (100.0)	20 (24.4)	11 (13.4)	40 (48.8)	11 (13.4)	
Missing	6	4	0	1	1	
One-hour OGTT						
<8.9	112 (100.0)	62 (55.4)	15 (13.4)	25 (22.3)	10 (8.9)	0.019
8.9-<10.0	109 (100.0)	56 (51.4)	16 (14.7)	32 (29.4)	4 (3.7)	
10.0-<10.8	135 (100.0)	71 (52.6)	23 (17.0)	28 (20.7)	13 (9.6)	
10.8+	123 (100.0)	43 (35.0)	19 (15.4)	48 (39.0)	12 (9.8)	
Missing	175	97	6	61	11	
Two-hour OGTT						
<8.0	150 (100.0)	55 (36.7)	33 (22.0)	48 (32.0)	14 (9.3)	<0.001
8.0-<8.7	170 (100.0)	104 (61.2)	13 (7.6)	43 (25.3)	9 (5.3)	
8.7-<9.4	158 (100.0)	86 (54.4)	12 (7.6)	51 (32.3)	9 (5.7)	
9.4+	169 (100.0)	82 (48.5)	20 (11.8)	52 (30.8)	14 (8.3)	
Missing	7	2	1	0	4	
Combined diagnosis categories						
Fasting and 1 h or 2 h	91 (100.0)	20 (22.0)	13 (14.3)	43 (47.3)	15 (16.5)	<.0001
Fasting only	77 (100.0)	21 (27.3)	18 (23.4)	27 (35.1)	11 (14.3)	
1 h and 2 h only	113 (100.0)	56 (49.6)	17 (15.0)	31 (27.4)	8 (7.1)	
1 h only	78 (100.0)	44 (56.4)	13 (16.7)	15 (19.2)	6 (7.7)	
2 h only	295 (100.0)	188 (63.7)	18 (6.1)	78 (26.4)	10 (3.4)	

**Table 4 nutrients-15-01226-t004:** GDM diagnostic group and perinatal outcomes.

Outcome	Total	Fasting and 1 h or 2 h	Fasting only	1 h and 2 h only	1 h only	2 h only	Chi Squared *p* Value
	*n* = 654	*n* = 91	*n* = 77	*n* = 113	*n* = 78	*n* = 295	
Gestational age at diagnosis							
<20	129 (19.7)	14 (15.4)	18 (23.4)	24 (21.2)	14 (17.9)	59 (20.0)	0.939
20–27	210 (32.1)	31 (34.1)	28 (36.4)	36 (31.9)	26 (33.3)	89 (30.2)	
28–32	269 (41.1)	40 (44.0)	29 (37.7)	43 (38.1)	33 (42.3)	124 (42.0)	
33+	41 (6.3)	5 (5.5)	2 (2.6)	8 (7.1)	5 (6.4)	21 (7.1)	
Missing	5 (0.8)	1 (1.1)	0 (0.0)	2 (1.8)	0 (0.0)	2 (0.7)	
Gestational age at treatment							
<20	80 (12.2)	11 (12.1)	8 (10.4)	16 (14.2)	8 (10.3)	37 (12.5)	0.911
20–27	167 (25.5)	16 (17.6)	20 (26.0)	28 (24.8)	20 (25.6)	83 (28.1)	
28–32	307 (46.9)	48 (52.7)	39 (50.6)	49 (43.4)	38 (48.7)	133 (45.1)	
33+	95 (14.5)	15 (16.5)	10 (13.0)	18 (15.9)	12 (15.4)	40 (13.6)	
Missing	5 (0.8)	1 (1.1)	0 (0.0)	2 (1.8)	0 (0.0)	2 (0.7)	
Gestational age at delivery							
<37	48 (7.3)	4 (4.4)	4 (5.2)	11 (9.7)	6 (7.7)	23 (7.8)	0.031
37	63 (9.6)	17 (18.7)	5 (6.5)	8 (7.1)	6 (7.7)	27 (9.2)	
38	176 (26.9)	31 (34.1)	21 (27.3)	29 (25.7)	27 (34.6)	68 (23.1)	
39	271 (41.4)	35 (38.5)	38 (49.4)	47 (41.6)	26 (33.3)	125 (42.4)	
40	86 (13.1)	3 (3.3)	8 (10.4)	18 (15.9)	12 (15.4)	45 (15.3)	
41+	8 (1.2)	0 (0.0)	1 (1.3)	0 (0.0)	1 (1.3)	6 (2.0)	
Missing	2 (0.3)	1 (1.1)	0 (0.0)	0 (0.0)	0 (0.0)	1 (0.3)	
Mode of delivery							
Normal vaginal	322 (49.2)	39 (42.9)	40 (51.9)	52 (46.0)	37 (47.4)	154 (52.2)	0.469
Instrumental	98 (15.0)	16 (17.6)	11 (14.3)	15 (13.3)	10 (12.8)	46 (15.6)	
Planned CS	199 (30.4)	31 (34.1)	26 (33.8)	39 (34.5)	27 (34.6)	76 (25.8)	
Emergency CS	34 (5.2)	4 (4.4)	0 (0.0)	7 (6.2)	4 (5.1)	19 (6.4)	
Missing	1 (0.2)	1 (1.1)	0 (0.0)	0 (0.0)	0 (0.0)	0 (0.0)	
Small for gestational age	72 (11.0)	8 (8.8)	7 (9.1)	12 (10.6)	11 (14.1)	34 (11.5)	0.814
Large for gestational age	63 (9.6)	12 (13.2)	11 (14.3)	5 (4.4)	6 (7.7)	29 (9.8)	0.126
Macrosomia (>4 kg)	39 (6.0)	5 (5.5)	9 (11.7)	4 (3.5)	4 (5.1)	17 (5.8)	0.212
Pre-eclampsia	11 (1.7)	1 (1.1)	2 (2.6)	3 (2.7)	1 (1.3)	4 (1.4)	0.829
Pregnancy hypertension	33 (5.0)	7 (7.7)	4 (5.2)	5 (4.4)	7 (9.0)	10 (3.4)	0.229
Neonatal hypoglycaemia	55 (8.4)	11 (12.1)	9 (11.7)	11 (9.7)	2 (2.6)	22 (7.5)	0.145
Jaundice	48 (7.3)	6 (6.6)	3 (3.9)	11 (9.7)	5 (6.4)	23 (7.8)	0.637
Respiratory distress	50 (7.6)	7 (7.7)	8 (10.4)	11 (9.7)	4 (5.1)	20 (6.8)	0.639
Admitted to NICU	70 (10.7)	9 (9.9)	5 (6.5)	15 (13.3)	7 (9.0)	34 (11.5)	0.608

## Data Availability

Data is unavailable due to privacy restrictions.

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
