# Peer review of "Fasting Glucose Level on the Oral Glucose Tolerance Test Is Associated with the Need for Pharmacotherapy in Gestational Diabetes Mellitus"

_nutrients, 2023, doi:10.3390/nu15051226_

Round 1

Reviewer 1 Report

Thank you for inviting me to review this interesting paper. Comments below are intended to help improve it. 

Abstract - Not sure what this adds to the abstract - it seems very specific

Women with GDM are known to be less likely to birth at full term, with greater likelihood of obstetric intervention.

Worth stating in the abstract that this study covered two different criteria 

Why did 58% of women have a fetal ultrasound for growth in the third trimester? should this not be higher?

Mean GA at scan was 31-32 weeks - so all normally distributed? With such high missing numbers is this useful information?

Worth a more extensive review of the literature  - this has been shown before but you have not referenced other papers  other than ACHOIS

Author Response

Abstract - Not sure what this adds to the abstract - it seems very specific: Women with GDM are known to be less likely to birth at full term, with greater likelihood of obstetric intervention.

This line has been removed from the abstract.

Worth stating in the abstract that this study covered two different criteria 

This has been added to the abstract (changes highlighted).

Why did 58% of women have a fatal ultrasound for growth in the third trimester? should this not be higher?

There was some missing data sets, which largely incorporated women who had scans done outside the hospital setting. This was unable to be captured.

Mean GA at scan was 31-32 weeks - so all normally distributed? With such high missing numbers is this useful information?

In light of this reviewer’s comments and review 2 comments, the section on ultrasound has been moved to supplemental material.

Worth a more extensive review of the literature  - this has been shown before but you have not referenced other papers  other than ACHOIS

 Further references have been added (changes highlighted).

Reviewer 2 Report

This is a useful addition to the information linking management and outcomes in gestational diabetes mellitus (GDM). The choice of study years was complicated by change in diagnostic criteria implemented midway.  

There are some points which require clarification. Strictly the diagnosis of GDM is based on a OGTT undertaken at 24-28 weeks gestation. It is clear from tabulated data that a significant number of women had early OGTT. While this does not detract ultimately is may concern the purists and the place of early OGTT is still unclear.

The study is of a retrospective "cohort". Was this cohort ALL women attending the multidisciplinary clinic or a subset? If the latter what determined their selection?

In the comprehensive statistical analysis, the ultimate outcome for these patients and their babies - namely how big or small were the babies - is provided in terms of Small or Large for gestational age with macrosomia as a subset. Did pharmacotherapy make a difference?  I could not find that enumerated. Or were numbers too small overall to allow that to be considered? 

Table 4 (foetal ultrasound in 3rd trimester) does not add materially to the discussion as there were no differences. I suggest it be placed as Supplementary material if the authors feel it is of sufficient importance. 

Author Response

This is a useful addition to the information linking management and outcomes in gestational diabetes mellitus (GDM). The choice of study years was complicated by change in diagnostic criteria implemented midway.  

There are some points which require clarification. Strictly the diagnosis of GDM is based on a OGTT undertaken at 24-28 weeks gestation. It is clear from tabulated data that a significant number of women had early OGTT. While this does not detract ultimately is may concern the purists and the place of early OGTT is still unclear.

Thank you for this comment. The sample does contain early GDM diagnoses. This was discussed, and as, pointed out, was felt not to detract from the overall message of the paper. While the early OGTT is still somewhat unclear, clinical decisions are still being made using these results. Further, it provides a more wholistic representation of the clinical cohort, and so felt to add value.

The study is of a retrospective "cohort". Was this cohort ALL women attending the multidisciplinary clinic or a subset? If the latter what determined their selection?

This was not the entirety of the cohort, but rather a random selection.

In the comprehensive statistical analysis, the ultimate outcome for these patients and their babies - namely how big or small were the babies - is provided in terms of Small or Large for gestational age with macrosomia as a subset. Did pharmacotherapy make a difference?  I could not find that enumerated. Or were numbers too small overall to allow that to be considered? 

There was no difference in LGA or SGA. This is outlined in table 5 and the results, line 24 page 3. The likely reason for this is treatment effect, which is further discussed in the discussion.

Table 4 (foetal ultrasound in 3rd trimester) does not add materially to the discussion as there were no differences. I suggest it be placed as Supplementary material if the authors feel it is of sufficient importance. 

This has been moved to supplemental material, rather than the main paper, as per reviewer’s suggestions.